# Analyzing the Effects of Cr and Mo on the Pearlite Formation in Hypereutectoid Steel Using Experiments and Phase Field Numerical Simulations

**DOI:** 10.3390/ma17143538

**Published:** 2024-07-17

**Authors:** Faisal Qayyum, Ali Cheloee Darabi, Sergey Guk, Vinzenz Guski, Siegfried Schmauder, Ulrich Prahl

**Affiliations:** 1Institut für Metallformung, Technische Universität Bergakademie Freiberg, Bernhard von Cotta-Str. 4, 09599 Freiberg, Germany; sergey.guk@imf.tu-freiberg.de (S.G.); ulrich.prahl@imf.tu-freiberg.de (U.P.); 2Institut für Materialprüfung, Werkstoffkunde und Festigkeitslehre, Universität Stuttgart, Pfaffenwaldring 32, 70569 Stuttgart, Germany; ch.darabi@gmail.com (A.C.D.); vinzenz.guski@imwf.uni-stuttgart.de (V.G.); siegfried.schmauder@imwf.uni-stuttgart.de (S.S.)

**Keywords:** pearlitic steels, electron microscope, phase field simulations, image processing, pearlite morphology, statistical analysis

## Abstract

In this study, we quantitatively investigate the impact of 1.4 wt.% chromium and 1.4 wt.% molybdenum additions on pearlitic microstructure characteristics in 1 wt.% carbon steels. The study was carried out using a combination of experimental methods and phase field simulations. We utilized MatCalc v5.51 and JMatPro v12 to predict transformation behaviors, and electron microscopy for microstructural examination, focusing on pearlite morphology under varying thermal conditions. Phase field simulations were carried out using MICRESS v7.2 software and, informed by thermodynamic data from MatCalc v5.51 and the literature, were conducted to replicate pearlite formation, demonstrating a good agreement with the experimental observations. In this work, we introduced a semi-automatic reliable microstructural analysis method, quantifying features like lamella dimensions and spacing through image processing by Fiji ImageJ v1.54f. The introduction of Cr resulted in longer, thinner, and more homogeneously distributed cementite lamellae, while Mo led to shorter, thicker lamellae. Phase field simulations accurately predicted these trends and showed that alloying with Cr or Mo increases the density and circularity of the lamellae. Our results demonstrate that Cr stabilizes pearlite formation, promoting a uniform microstructure, whereas Mo affects the morphology without enhancing homogeneity. The phase field model, validated by experimental data, provides insights into the morphological changes induced by these alloying elements, supporting the optimization of steel processing conditions.

## 1. Introduction

Steels, which are one of the most widely used materials in various industries, possess a complex microstructure that can be customized through appropriate alloying and heat treatment methods [1]. Understanding the evolution of microstructures in steels is essential for the design and manufacturing materials with the desired microstructural attributes [2]. Various studies have shown that the different morphologies of pearlitic and spheroidized steels have a significant impact on mechanical properties, such as strength, hardness, toughness, and fatigue resistance [3,4,5]. In pearlitic steels, the lamellar structure provides high strength but limited ductility [6,7]. Resistance to fatigue and creep is also affected by the presence of pearlite, as the interfaces between ferrite and cementite act as stress concentration sites [8,9,10,11]. Alloying elements, such as carbon, chromium, and molybdenum, significantly influence the microstructural morphology of steel alloys [12,13]. Carbon, in particular, plays a critical role in determining the morphology of pearlitic and spheroidization degree [14] as it affects the spacing and thickness of the ferrite and cementite in pearlitic steels [15]. Higher carbon concentrations have been reported to lead to a finer pearlitic structure, resulting in increased strength and hardness [14], but such materials are also prone to faster carbide build up on the grain boundaries. On the other hand, the spheroidization of cementite particles in steel alloys is promoted by lower levels of carbon [16] with less carbide build up. In addition to carbon, chromium and molybdenum are commonly added as alloying elements to enhance the corrosion resistance and high temperature properties of steels [17]. For example, chromium promotes the formation of stable carbides [18], while molybdenum aids in the refinement of the microstructure and the improvement of high-temperature strength [19]. Recent work by several researchers has further improved our understanding of the role of alloying elements in pearlite morphology in carbon steels [20,21,22].

The speed of phase transformations in steels, including the developments of pearlitic morphology and spheroidization have a direct impact on the carbon footprint of the manufactured products [23,24]. In today’s environmentally conscious world, reducing energy consumption and greenhouse gas emissions are pressing concerns [25,26]. Thermomechanical treatments of alloys comprising sustainable elemental compositions offer a promising approach to achieve the desired microstructural attributes in steels with lower energy input and reduced environmental impact [25,27]. Optimizing thermal treatment parameters, including temperature, cooling rate, and holding time, can accelerate phase transformations and preserve desired microstructural features. This approach improves the properties of steel more efficiently and sustainably, reducing energy use. Although traditional experimental methods provide information about the effects of alloying on microstructure and properties, they face challenges in cost, time, and precision [28,29,30,31]. Moreover, the static nature of experiments makes it challenging to accurately capture and measure every step of the transformation process. These limitations have led to less deterministic and less repeatable experimental results. Although advances in in situ characterization techniques [10,32,33] are gradually improving our ability to observe and understand microstructural evolution in real time, offering insights into dynamic phase transformations [34,35], they still lag in cost and time domains.

Phase field simulations have revolutionized the field of materials science, enabling researchers to investigate the evolution of microstructures in a computationally efficient and realistic manner [36,37]. These simulations, based on the phase field method, simulate the dynamics of microstructural changes by representing the material system as a continuous field that evolves over time [38,39,40]. By solving the governing equations that describe the thermodynamic and kinetic behavior of the system, phase field simulations provide a comprehensive understanding of the spatial and temporal evolution of microstructures [41]. The phase field method is based on the concept of order parameters, which describe the phase composition and morphology within the material. The evolution of these order parameters is governed by a set of partial differential equations, such as the Cahn–Hilliard equation [42] or the Allen–Cahn equation [43], which incorporate the thermodynamic driving forces and the mobility of interfaces [44]. The key advantage of phase field simulations lies in their ability to capture complex phenomena, including phase transformations, interface motion, nucleation, growth, and coarsening, without the need for explicit tracking of interfaces or boundaries [45,46,47,48,49].

Amos et al. utilized phase field simulations to investigate the kinetics of pearlite transformation during annealing in steel [50]. Their study revealed the influence of temperature, alloying elements, and cooling rate on pearlite morphology, facilitating the understanding of microstructural evolution and mechanical properties. Similarly, Mecozzi et al. focused on phase field simulations to examine the effects of quenching parameters on the martensite formation of martensite in steel [51], shedding light on the microstructural changes and resulting hardness. Zhang et al. demonstrated that controlling the quenching rate can significantly affect the transformation kinetics and resultant microstructure in steel [52]. Their simulations revealed that a slower quenching rate led to the formation of coarser martensite structures with improved toughness, while a rapid quenching rate produced finer martensite structures with higher hardness. In all the discussed cases, various phase field modeling techniques utilizing material libraries were applied, drawing thermodynamic and kinetic parameters from experimental data and fine-tuning interface parameters to align with observed microstructures. Although these calibration approaches generally yield simulations consistent with experimental results, they need to be re-calibrated every time. For example, Amos et al. [50] successfully calibrated their phase field model using a material library approach to predict the formation of ferrite and pearlite during annealing in low carbon steels. In addition to material libraries, the adjustment of interface parameters has been utilized for model calibration. This approach involves fine-tuning the interface energy and mobility to reproduce the observed microstructural features. For instance, Zhang et al. [52] adjusted the interface parameters in their phase field simulations to accurately capture the formation and growth of martensite plates during quenching in steel alloys. Such calibration methods have demonstrated their capability to reproduce microstructures similar to the ones that are experimentally observed and PFM simulations provide valuable insights into the underlying mechanisms of how these observed microstructures form and evolve with changing parameters such as chemistry, temperature and time.

Phase field simulations for heat treatment processes in various types of steel have made notable advancements [53,54,55]; however, the dynamic prediction of microstructural changes, particularly the evolution of pearlite morphology during cooling from a fully austenitic phase, remains a challenge. Specifically, while the impact of alloying elements, such as Cr and molybdenum (Mo), on the final microstructure has been studied, their effect on the dynamic evolution of pearlite morphology is not fully understood. This is of considerable importance due to the implications it has on the mechanical behavior of material. Existing models require labor-intensive experimental procedures for calibration, limiting their efficiency and commercial applicability [56,57]. Furthermore, the use of software tools, such as MatCalc, to derive thermodynamic parameters from the relevant literature, which is an important step in these simulations, has yet to be properly used.

To address this, the study sets three primary objectives. First, the objective is to develop a numerical simulation method capable of dynamically predicting the evolution of pearlite morphology in binary and tertiary carbon steels. This involves an understanding of how alloying elements such as Cr and Mo influence the pearlitic transformation during the cooling process. Second, the study will use experimental measurements to obtain critical material data, and calibrate interface fitting parameters, thereby enhancing the reliability of the simulation model. Third, develop and use a robust microstructural feature characterization technique to quantitatively analyze and compare pearlite morphology across different alloys and heat treatment procedures from experiments and numerical simulation predictions. This three-pronged approach aims to establish a more efficient methodology for predicting and optimizing heat treatment processes in various types of steel.

## 2. Experimental and Phase Field Simulation Methodologies

### 2.1. Sample Preparation

The available VIM LAB 20–50 (SECO WARWICK EUROPE (subsidiary of Retech Systems LLC, Ukiah, CA, USA)) vacuum induction setup was used to cast billets with the chemical compositions provided in Table 1. Each alloy is assigned a respective acronym that will be used throughout the document for reference.

The cast billets were rolled into a wire using multipass rolling starting at 1250 °C and ending at ~850 °C. The rolling experiments were conducted using a Trio-Walzgerüst (Freiberg, Germany), a versatile rolling mill designed to accommodate various groove sequences and commonly used for fundamental studies on material flow during caliber rolling. The device had roll diameters ranging from 280 to 320 mm, a maximum rolling steep of 8 m/s with maximum initial block dimensions of 48 × 48 × 2500 mm, and can produce wires from Φ 6 to 16 mm. The Trio was used to roll 12 mm diameter wire that was then cut into manageable lengths and homogenized at 1250 °C for 8 h. Solid and hollow cylindrical samples 10 mm in length and a 5 mm outer diameter were manufactured from all reference materials for the dilatometer tests.

### 2.2. Analysis of Phase Transformation Behavior

To investigate the behavior of phase transformations of the materials, the tests were carried out on DIL 805-A by Bähr (TA Instruments, New Castle, DE, USA). During these tests, the samples under vacuum were heated by induction at 1 °C/s with a heating rate of up to 1200 °C, and held there for 5 min and then cooled using nitrogen at 1 °C/s down to room temperature with 5 data points per degree centigrade change recorded. The temperature dilation curves shown in Figure 1 are plotted and analyzed to obtain key phase transformation temperatures, as presented in Table 2. It is evident that, with the addition of chromium and molybdenum, the temperature gap between heating and cooling increases.

The phase transformation is a dynamic process that largely depends on the heating and cooling rates and, for that, usually TTT curves are constructed. In the current work, the TTT curves were calculated using J-Mat Pro v12 with its steel database. More details about the model, database, and methodology are published elsewhere [58]. The resulting TTT curves for the 1C, 1.4Cr, and 1.4Mo alloys are presented in Figure 2. These curves were later used to adjust experimental parameters to obtain a fully pearlitic microstructure in all three alloys and diffusion coefficients for the phase field calculations.

It can be observed from the presented TTT diagrams that the rate of transformation in Fe-1C is very high, with a greater tendency toward pearlite transformation (as shown in Figure 2a). As chromium is added to the alloy, the transformation is delayed but the tendency of the bainitic transformation becomes higher (as shown in Figure 2b). As the molybdenum is added to the reference alloy, the transformation is slowed further with a much greater tendency toward bainitic transformation (as shown in Figure 2c). The DIL tests were carried out at 1 °C/s heating and cooling rates. The green, round dots on the TTT diagrams mark the start of the transformation, and black dots represent the more than 90% completion of the transformation as analyzed from the dilatometer data analysis, without looking at the microstructure of the presented state.

### 2.3. Modeling and Assumptions

To predict microstructure evolution, the phase field method (PFM), as proposed by Steinbach et al. [59], was utilized. In this method, the phase is known by the parameter *φ*, which defines the type of phase in the regions and changes between 0 and 1 during the simulation. The local fraction of phases in a grain is represented by the parameter $\varphi_{\alpha }$, known as the sum of local phase fractions, which is equal to 1 ($\sum \varphi_{\alpha }=1$). The rate of these parameters is introduced as Equation (1) and applied to calculate the changes in phase *α* based on the difference between the thermodynamic properties of the neighboring phases *β* and *γ* during the process [60]:(1)$${\dot{\varphi }}_{\alpha }=\sum_{\alpha \neq \beta }^{n}M_{\alpha \beta }^{\varphi }[b∆G_{\alpha \beta }-\sigma_{\alpha \beta }\left(K_{\alpha \beta }+A_{\alpha \beta }\right)+\sum_{\alpha \neq \beta \neq \gamma }^{\upsilon }j_{\alpha \beta \gamma }]$$

The movement of the interface between the *α* and *β* phases is calculated using the expressed function in the brackets. The parameters $∆G_{\alpha \beta }$ and $K_{\alpha \beta }$ illustrate the difference between the *α* and *β* phases in the Gibbs free energy and pairwise curvature calculated according to Equation (2) and Equation (3), respectively. The parameter b is a pre-factor and $J_{\alpha \beta \gamma }$ refers to the triple transition between the α, β, and γ phases, as shown in Equation (4) and Equation (5), respectively:(2)$$∆G_{\alpha \beta }=\frac{1}{\nu^{m}}(\mu_{\beta }^{0}-\mu_{\alpha }^{0})$$
(3)$$K_{\alpha \beta }=\frac{\pi^{2}}{2\eta^{2}}\left(\varphi_{\beta }-\varphi_{\alpha }\right)+\frac{1}{2}(\nabla^{2}\varphi_{\beta }-\nabla^{2}\varphi_{\alpha })$$
(4)$$b=\frac{\pi }{\eta }(\varphi_{\alpha }+\varphi_{\beta })\sqrt{\varphi_{\alpha }\varphi_{\beta }}$$
(5)$$J_{\alpha \beta \gamma }=\frac{1}{2}(\sigma_{\beta \gamma }-\sigma_{\alpha \gamma })(\frac{\pi^{2}}{\eta^{2}}\varphi_{\gamma }+\nabla^{2}\varphi_{\gamma })$$

The parameter $M_{\alpha \beta }^{\varphi }$ in Equation (1) is the mobility between phases α and β, which is calculated according to Equation (6) as a function of the kinetic coefficient in the Gibbs–Thomson equation ($\mu_{\alpha \beta }^{G}$):(6)$$M_{\alpha \beta }^{\varphi }=\frac{\mu_{\alpha \beta }^{G}}{1+\frac{\mu_{\alpha \beta }^{G}\eta ∆s_{\alpha \beta }}{8}\left\{\sum_{i}m_{i}^{l}\sum_{i}\left[{\left(D_{\alpha }^{ij}\right)}^{-1}(1-k_{j})c_{j\alpha }\right]\right\}}$$
where *i* and *j* represent the phases in the model. The parameter $∆s_{\alpha \beta }$ is the difference in entropy, and *η* is the thickness of the interface between the phases. The parameter $D_{\alpha }^{ij}$ presents the diffusion matrix of phases *i* and *j* in phase α. Also, the parameters $m_{i}^{l}$ and $k_{j}$ refer to the liquidus line slop and the partition coefficient, respectively.

### 2.4. Identification and Calibration of PFM Simulation Parameters

During the calibration of PFM simulations, the heat treatment routines to achieve a pearlitic microstructure for three low-alloy steels discussed previously in Section 2 are simulated using MICRESS v7.2 software [37]. Since all three materials were heat treated to achieve a fully austenitic microstructure, the PFM simulations are initialized with a fully austenitic steel of the respective chemical compositions presented in Table 1. The grain size is calculated using MatCalc v5.51 software [4] to generate the initial microstructure based on the experimental observations. For the simulation, a “phase concentration” model in MICRESS v7.2 and periodic boundary conditions (called PPPP in MICRESS) are utilized.

For the modeling of the reference 1C alloy, the binary Fe-C phase diagrams were taken from MatCalc v5.51 software and used for the simulation. Due to the fact that different heat treatment conditions are predicted in this study and the Gibbs free energy and the mobilities between phases are strongly dependent on temperature, these two values are considered as a function of temperature. Figure 3 shows the change in Gibbs free energy as a function of temperature in each phase of the three materials, which were calculated using MatCalc v5.51 software. The mobilities have been assumed to obey the Arrhenius function provided in Equation (7) [61].
(7)$$M_{ij}=M_{ij}^{0}\;.\;exp(\frac{Q_{ij}^{M}}{RT})$$
where $M_{ij}^{0}$ is the mobility pre-factor between phases *i* and *j*, $Q_{ij}^{M}$ is the activation energy, and *T* and *R* are the temperature and the ideal gas constant, respectively.

Various values have been reported in the literature for the mobility between different phases. Therefore, the pre-factor parameters between the different phases are used here for calibration by comparing the experimental and PFM simulation results. Here, the activation energy was considered as a constant value, 140 KJ/mol, according to previously published work [62,63]. Table 3 shows the other thermodynamic parameters used in this study for the three materials. Here, the interfacial energies between the phases were taken from the literature [64,65]. Carbon diffusion is also temperature dependent and is defined by the initial coefficient and activation energy based on the Arrhenius equation, but its dependence on composition has been neglected for all three materials, and these values are presented in Table 3.

Figure 4 shows the qualitative influence of mobility between the ferrite and austenite phase areas ($M_{\alpha \gamma }$) on the predicted microstructure. The results show that, as $M_{\alpha \gamma }$ increases, the inter lamella spacing increases, which means that the areas of the ferrite phase grow faster, and the cementite phase has less time to grow. Moreover, as $M_{\alpha \gamma }$ decreases, the inter lamella spacing decreases and the width of the lamella also increases, which means that the cementite phase areas have more time to grow in the ferrite and austenite phase areas.

Figure 5 shows the influence of $M_{\gamma \theta }$ on the predicted microstructures. The results show that, when this mobility increases, the cementite phase areas grow much faster than the ferrite phase areas, and since the growth of these two phases is not consistent, the cementite phase areas cannot follow a lamellar growth, which deviates from its own pattern and follows an irregular branch-type growth in the austenitic phase areas. On the other hand, if it decreases less than $M_{\alpha \gamma }$, the cementite phase areas do not have enough time to grow in parallel with the ferrite phase areas, so the cementite phase areas just begin to grow at the boundary with the ferrite phase areas, and the microstructure cannot have a lamellar structure.

The influence of $M_{\alpha \theta }$ on the predicted microstructure is shown in Figure 6. Since the mobility between ferrite and cementite is much less than that of the other two mobilities (with austenite), it could be expected that it will have less influence on the microstructures. However, the results show that this mobility has an important influence on the fragmentation of the lamellar phases. Therefore, the lamellar phases start to fragment sooner when $M_{\alpha \theta }$ increases in comparison to a lower value.

To verify the PFM simulation based on the experimental results, these parameters were calibrated with experimental observations at different temperatures and are presented in Table 4 for the three materials. The results are shown in Figure 7—there is good qualitative agreement between the experimental and PFM results. These calibrated data are used for the prediction of microstructures under different heat treatment conditions, which are presented in the next section.

### 2.5. Microstructural Analysis and Feature Characterization

After appropriate heat treatments, cylindrical experimental samples were sectioned from the middle horizontally and vertically, embedded in epoxy, ground, and polished with successively decreasing grit sizes from 800 µm to 1 µm, and finally vibration polished for 4 h. The polished samples were then etched with a 5% Nital solution for 8–10 s (depending on the amount of cementite present) to create a clear contrast between the cementite and the ferrite lamella in the pearlitic microstructure. The samples were then analyzed under an electron microscope at an appropriate magnification with an adequate current, voltage, brightness, and contrast settings to obtain clear pictures of the pearlite colonies perpendicular to the observation surface. It is important to note that the magnification of each SEM image was set differently depending on the size and resolution of the observed pearlite colony of interest to record a clear image. The SEM images were then used to identify the microstructural features. Then the simulation predictions were directly saved during post-processing and used for further analysis. The microstructural features of experimental and numerical simulation results were identified measures and analyzed semiautomatically using FIJI ImageJ. The details of the measuring method are provided in Appendix A.

Lamella morphology is quantitatively characterized from the SEM images. The probability distribution functions (PDFs) are used to plot the data sets and develop distribution plots to understand the variability in the data. The means of the 1st and 3rd quartiles were calculated from these distribution plots to capture the variance and scatter in the data set. All the data sets for the case of 1C with varying holding temperatures are presented in Appendix A for reference; for all cases, the evolution of the microstructural features is presented in this section.

The scatter in data related to cementite lamella morphology is a reflection of the complex and variable nature of pearlitic steels. The method described in Appendix A was employed, including multiple characterizations and selective measurements, to capture a representative picture of the microstructure. Although this approach may result in some scatter, it ensures that the data provide a comprehensive understanding of the morphological attributes of interest. To gain a complete and meaningful interpretation of such data, it is essential to consider the corresponding microstructural figures alongside the statistical analysis. Therefore, the readers are encouraged to look into the raw microstructural data set provided in Appendix B for each case to co-relate the quantitative morphology parameters with a visual outlook for a better understanding.

## 3. Results

The aim of this study is to experimentally understand and model the pearlitic transformation in each alloy considered and compare the outcomes with the reference alloy to develop an understanding of how the addition of 1.4 wt.% Cr or Mo to the reference alloy affects pearlite formation and morphology. In this section, the effect of holding temperature on simulations and experiments is discussed, and then, for the selected holding temperatures, the effect of holding time is investigated and compared.

### 3.1. Effect of Holding Temperature on Pearlite Morphology

It is visually observed in Figure 8 that lamella morphology becomes more separated and has a wider distribution of shapes with increasing temperature both in the experiments and numerical predictions. The results show that at 530 °C, the cementite particles are small and cannot grow into lamellar structures. However, as the temperature increases, there are more defined lamellar structures. After 600 °C, there seem to be only a few cementite particles, and as the temperature increases, the spacing and the thickness of the lamellae increase. Parametric distributions are simplified to show the evolution of the mean and data variance with increasing holding temperatures shown in Figure 9. The quantitative analysis provides a deeper understanding of the morphologies and their comparisons.

The experimental and numerical results are compared in Figure 9. The experimental results indicate that lamellar length increases with holding temperature, with measurements ranging from 300 nm to over 600 nm. The simulations correspondingly show a strong positive correlation, with lamellar lengths increasing from 150 nm at 530 °C to 850 nm at 660 °C. This trend could be attributed to the increase in the carbon diffusion coefficient with temperature. At higher temperatures, carbon atoms have more kinetic energy and can diffuse more rapidly within the microstructure, resulting in the more extensive growth of the cementite and ferrite phase areas. Thus, the lamellae become longer as the atoms have more freedom to migrate and occupy different positions.

On the contrary, lamellar thickness varies less predictably with temperature; experimentally, it is in the range of 70–150 nm at lower temperatures and 100–170 nm at the highest temperature observed. Simulation data also suggest an increase in a nonlinear fashion, from approximately 95 nm at 530 °C to 200 nm at 660 °C. The increase in thickness may be attributed to enhanced carbon diffusion at higher temperatures, which promotes the formation of broader cementite lamellae. 

Circularity, a metric of the lamellae’s shape regularity, tends to increase with temperature in the experimental findings for the 1C alloy, while the simulation indicates a decrease, suggesting irregular shapes at higher temperatures due to rapid carbon diffusion. This trend suggests that, as the temperature increases, the lamellae, possibly due to the accelerated diffusion of carbon at higher temperatures, rapidly break into globules. Finally, the experimental measurement of the interlamellar spacing shows complex temperature dependence, with a notable increase at lower and higher temperatures, but a decrease was observed at an intermediate temperature of 580 °C, partially supported by simulations indicating a decrease between 580 °C and 600 °C. The overall increase in spacing may be attributed to the faster diffusion of carbon at higher temperatures, which can result in larger distances between the lamellae. However, the local decrease could be actual and arise from variations in the microstructure and heterogeneities in the carbon distribution.

The effect of chromium on pearlite lamella morphology is analyzed by systematic sample preparation and quantitative microstructure characterization. The microstructural image presented in Figure 10 demonstrates that, at lower temperatures, carbon is concentrated on the grain boundaries, resulting in a poorly defined lamella. As the temperature increases, the diffusion coefficient of carbon increases, leading to an increasingly defined formation of pearlite lamella, as observed in Figure 10. The distribution plots of the attributes in Appendix B show a trend of longer lamellar length, narrower lamella thickness, improved circularity, and homogeneous interlamellar spacing.

When the morphological evolution of pearlite in the 1.4Cr alloy is resolved as a function of holding temperature, interesting trends are observed in Figure 11. Chromium’s introduction distinctly influences pearlite morphology with a heavy segregation of carbides on the grain boundaries, apparent in the reduced lamellar thickness predominantly below 70 nanometers at 560 °C, exhibiting high variance at lower temperatures and stabilizing with increased temperature. In parallel, the experimental lamellar length remains consistent, peaking at 690 °C. Simultaneously, circularity peaks at 560 °C and 619 °C, with optimal ratios at 610 °C, suggesting selective growth dynamics. A key observation is the minimal interlamellar spacing at lower temperatures, which progressively expands with increasing temperature, implying the homogenization of the microstructure due to the influence of chromium.

When experimental trends with simulations are observed, these phenomena are interpreted as the effect of decelerating the austenite-to-pearlite transformation, evident in the low density of pearlite at 530 °C, indicative of a transformation in its nascency, with growth restricted by sluggish diffusion. The higher holding temperatures result in elongated, but thinner, lamellae, due to the elevated diffusion rates, leading to decreased circularity, reflecting more extended morphologies. Interestingly, at 610 °C, the lamellae exhibit reduced length and thickness, suggesting temperature-induced fragmentation, but with a higher circularity indicative of a more isotropic shape. Further temperature elevation causes renewed growth in lamellar length, but thinning in profile, hinting at temperature-assisted elongation of the lamellae. At 690 °C, the lamellae are observed to be the longest, maintaining thinness and exhibiting a marked decrease in circularity, denoting irregular growth.

The temperature-dependent interlamellar spacing widening aligns with the lengthening and thinning of the lamellae, offering an insight into the diffusion behavior of carbon and chromium under thermal influence. Therefore, the addition of chromium to the 1C alloy appears to refine the morphology of the pearlite and stabilize the microstructural characteristics, resulting in a more homogenized and enhanced pearlite configuration within the hypereutectoid steel matrix.

When analyzing the effect of the 1.4% molybdenum addition on the 1C reference alloy, the samples were held at varying temperatures for 90 min. The qualitative results of the experiments and simulations are presented in Figure 12. From the high-contrast microstructural images in Figure 12, it is visible that, with increasing the temperature, the thickness of the lamella continuously increases while the length of the lamella decreases or stays constant. There is a large variance in the morphology, some lamella being very long, while others are small and circular. At lower temperatures, the pearlite microstructure is dominated by relatively short lamellae and small particles. This illustrates a stage in the pearlitic transformation, where the ferrite and cementite phase areas have not yet fully interacted. However, as the holding temperature increases, there is an obvious change in microstructural dynamics. The cementite particles gradually dissolve into the ferrite matrix, leading to an increase in the length of the lamellar phases. During the same time, the interlamellar spacing (ILS) also widens, a trend that continues with further increases in temperature. Moreover, a prolonged holding time of 90 min promotes a significant increase in the thickness of the lamellar phases.

In the context of pearlite morphology in the 1.4Mo alloy, with a holding time of 90 min, the addition of molybdenum introduces notable heterogeneity. The results of the experimental and numerical simulations for the quantitative comparison of the lamellar morphology are presented in Figure 13, which exhibits a pronounced variation in thickness and length. The circularity distribution, as shown, is expansive because of the onset of spheroidization within the microstructure. Interlamellar spacing (ILS) similarly displays a substantial increase with temperature, coupled with a broad variance at elevated temperatures, indicative of a pronounced anomaly in the microstructure introduced by molybdenum. Circularity is observed to be minimal at 550 °C and 635 °C, with a peak at 590 °C, reflecting an intricate interplay of microstructural dynamics throughout the temperature spectrum.

Simulations reveal a comparable upward trend in the mean length of lamellae with temperature, though a subsequent reduction is observed at higher temperatures, implying a limit to the elongation effect induced by temperature. The mean lamellar thickness also increases with temperature, denoting a trend toward more substantial lamellar growth at higher temperatures. However, circularity exhibits an inverse trend, diminishing with temperature and suggesting a shift toward more elongated lamellae. ILS initially increases, peaks, and then decreases at the highest temperature studied, pointing to a nonlinear response of the dissolution of cementite and lamellar distribution at varying thermal conditions.

The overall experimental and simulated data depict molybdenum’s impact on the 1.4Mo alloy, emphasizing the alloy’s propensity for higher circularity and significant variability in lamellar characteristics. This response underscores the role that molybdenum plays in pearlite spheroidization and growth behaviors, especially in the context of extended heat treatment times, which promote such transformations.

### 3.2. Effect of Holding Time on Pearlite Morphology

From this data set, an appropriate holding temperature of 580 °C was selected for further heat treatments with varying holding times to further calibrate the conditions to obtain the full pearlitic microstructure. Based on the TTT data of the reference alloy, the holding of the samples at 580 °C for various holding times of 1 min, 3 min 4 min, 5 min, and 30 min was investigated. The microstructural images presented in Figure 14 clearly show that, in experiments and in phase field numerical simulations, with increasing holding times, interlamellar spacing and lamellar thickness increase initially and then decrease with low variance in the data at a 30 min holding time. The length of the lamella also shows a similar trend, initially increasing slightly and then decreasing after 5 min.

The quantitative influence of the holding time on the microstructural evolution of the 1C alloy is determined at a holding temperature of 580 °C. The micrographs shown in Figure 14 are quantitatively analyzed and the data are presented in Figure 15, demonstrating the interplay between time, temperature, and pearlite lamellar characteristics. It is apparent that pearlite formation is a dynamic and rapid process. As early as 1 min into the holding time, lamellar structures begin to form with a mean length of approximately 840 nm. However, at this stage, the density of the lamellae is somewhat low, with a mean thickness of approximately 127 nm, and the circularity measure of about 0.41 suggests an irregular shape.

The circularity of the sample, combining the thickness and length parameters, has a mean value of around 0.4, with slightly higher middle and lower values at 3 min and holding times of 5 min and 30 min. Interlamellar spacing also becomes more homogeneous with increasing holding times, with less variance in the data. Figure 15 shows that lamella morphology remains similar with a slight drop in lamella circularity and ILS at longer holding times. The variance in the data also decreases generally with longer holding times, and therefore, it is safe to say that the microstructure is more homogeneous.

As the holding time extends to 5 min, lamellar density increases, although with a slight reduction in the mean length to approximately 780 nm. Interestingly, there is a marginal increase in the mean thickness, reaching approximately 130 nm, suggesting that the lamellae become denser during this period. At the 30 min mark, there is a considerable increase in both the lamellar length and thickness, reaching approximately 1000 and 160 nm, respectively. This suggests that, over time, pearlite lamellae grow in length, but also in thickness, resulting in a bulky carbide formation. The circularity measure decreases to about 0.37, indicating that the lamellae are becoming less circular and more irregular as time progresses.

Finally, at 60 min, the mean length and thickness reach their maximum values of approximately 1120 nm and 175 nm, respectively. Circularity also decreases to about 0.34, highlighting the increasingly irregular shape of the lamellae. Additionally, the interlamellar spacing, which indicates the mean distance between adjacent pearlite lamellae, generally increases from about 380 nm at 1 min to approximately 450 nm at 60 min. This trend implies that, as the lamellae grow in length and thickness over time, they also start to spread out more, possibly due to the increased diffusion of carbon at elevated temperature. Simultaneously, the transformation of the lamellar phases leads to the formation of cementite particles, thereby increasing the complexity of the microstructure.

After analyzing the effect of temperature, the effect of chromium on pearlite morphology at a holding temperature of 650 °C is analyzed with varying holding times of 5, 10, 15, and 60 min. The microstructural images in Figure 16 show that, for these holding times, a fully pearlite morphology is observed throughout the sample, with hints of ongoing spheroidization as the holding time increases. Quantitative analysis of the morphology of the cementite lamella is shown in Figure 17. 

The thickness of the lamella is confined at 30–70 nanometers for all holding times, which is finer than that observed for the reference alloy. The length of the lamella is in the range of 200–600 nm, with a relatively wider variance, but a mean value of 200–400 nm is observed that is the highest for the 10 min holding time, after which spheroidization starts to take place that breaks the lamella into smaller lengths. When the length and thickness parameters are combined with the circularity parameter, as shown in Figure 17, it is observed that the circularity is confined to 0.3, which increases slightly with the increasing holding time, and reaches its highest value at 60 min with larger variance in the data. This indicates that, with a longer holding time, spheroidization starts to occur in the sample. The ILS is plotted, which generally has a small variance in the data across all holding times, with values in the range of 40–100 nm, increasing with time and then slowly decreasing at 60 min, which indicates increasing microstructural fineness. Compared to the reference alloy, the addition of 1.4% chromium significantly improves the stability of pearlite lamella formation and results in a more homogeneous morphology throughout the microstructure. The holding time has a small effect on the ILS and lamella thickness, whereas, due to the breakage of the lamella at longer holding times, the circularity factor increases.

The PFM simulation micrographs reveal that the pearlite phase rapidly forms within the first 5 min. This early establishment of a well-defined pearlitic structure can be attributed to the efficient diffusion of carbon and chromium atoms at a given temperature, promoting the transformation of austenite to pearlite. As the holding time increases from 5 to 60 min, the micrographs show progressive changes in pearlitic morphology. The cementite lamellae begin to dissolve into the ferrite matrix, indicating that the system is moving toward a more thermodynamically stable state. This dissolution leads to an increase in interlamellar spacing (ILS), which is indicative of the gradual dispersion of lamellae within the ferrite matrix.

The data provided support these microstructural observations. After a 5 min holding time, the mean length of the cementite lamellae is approximately 1300 nm, with a relatively thin mean thickness of about 75 nm. The circularity value is around 0.14, suggesting that the lamellae exhibit a largely elongated shape, likely due to their rapid formation and subsequent dissolution. As the holding time extends to 10 min, the mean lamellar length decreases to around 758 nm, reflecting the ongoing dissolution of the cementite phase. Despite this, the lamellar thickness shows a slight increase to 77 nm, and the circularity measure also increases to 0.24, suggesting that the remaining lamellae are becoming thicker and more regular in shape. With a further increase in the holding time to 15 min, the mean lamellar length decreases further to approximately 610 nm, indicating continued dissolution. However, the thickness increases to about 84 nm and the circularity value increases to 0.30, implying that the remaining lamellae are still becoming thicker and more regular in shape.

At 30, 45, and 60 min holding times, the mean length continues to decrease to about 551, 529, and 514 nm, respectively. These values reflect the ongoing fragmentation and dissolution of the lamellae. During the course of the study, the thickness increases to 99, 109, and 115 nm, suggesting that the remaining lamellae are becoming significantly thicker over time. Circularity also increases to around 0.35, 0.38, and 0.40, indicating that the lamellae are growing more regular in shape. The interlamellar spacing, which is a measure of the average distance between adjacent lamellae, also shows an increase from about 59 nm at 5 min to approximately 198 nm at 60 min. This trend underscores the increasing dispersion of the lamellae within the ferrite matrix over time.

Although it is challenging to select a viable temperature for the quantitative analysis presented in Figure 18, for 1.4Mo, it was set at 550 °C and the holding time was varied between 10 and 90 min to observe the evolution of the pearlite lamella’s morphology. Data are analyzed using variance plots for thickness, length, and interlamellar spacing. The visualizations show that, at the shortest holding time of 10 min, the microstructure exhibits the emergence of shallow lamellar phases, indicating the onset of a pearlitic transformation. This phenomenon continues as the holding time increases, with an evident enhancement in the density of lamellar phases. The delayed start of pearlite formation, in comparison to other materials, can be attributed to the diffusion rates of carbon and molybdenum in the iron matrix. Thermodynamically, these rates govern the onset of the phase transformation, and in the case of the 1.4Mo alloy, the larger atomic size and lower diffusivity of molybdenum likely slow down the initial formation of pearlite. Furthermore, with an increased holding time, the cementite particles within the ferrite matrix gradually dissolve, which is a direct consequence of thermodynamic stability. At 550 °C, the system strives to achieve a lower energy state, resulting in the dissolution of metastable cementite particles and their subsequent incorporation into the more stable ferrite phase. This dissolution is concurrently accompanied by an increase in the fragmentation of the lamellar phases, which leads to an increase in the density of interlamellar spaces.

The qualitative assessment is supported by the quantitative morphological data plotted in Figure 18(b1–b3). At the holding temperature of 550 °C, the results in Figure 19 indicate that the thickness of the lamella is smaller (30 to 100 nm) at a 20 min holding time, which starts to increase continuously with the increasing holding time and reaches 70 to 100 nm at 90 min. The length of the lamella is shorter at a 10 min holding time, but increases significantly at a 90 min holding time. The circularity of the lamella is relatively high, ranging from 0.1 to 0.6. Interlamellar spacing is the least at a 20 min holding time, then continuously increases up to a 90 min holding time. The addition of molybdenum reduces the interlamellar spacing, the length of the lamella, and the variance in the data. However, the circularity of the lamella remains in a range similar to that of the reference alloy with slightly lower values. In general, the holding time affects the pearlite morphology of each alloy, but not as significantly as compared to the holding temperature. PFM simulation results also show similar results.

The mean length of the lamellae shows a general increase with time, from 615 nm at 10 min to approximately 971 nm at 120 min. This trend is indicative of the continued growth and maturation of the lamellar phases with time. The mean thickness of the lamellae also shows a steady increase with time, from 88 nm at 10 min to 212 nm at 120 min. This trend implies that the lamellae increase in thickness as more cementite dissolves in the ferrite matrix and deposits on the existing lamellae. On the contrary, the circularity, which starts at 0.30 at 10 min, increases initially, reaching its peak of 0.40 at 60 min, before gradually reducing to 0.38 at 120 min. This trend demonstrates that the lamellae initially become less circular and more elongated with time, after which they start to become more fragmented and thus more circular. The interlamellar spacing (ILS) shows a somewhat varied trend, initially decreasing from 531 nm at 10 min to about 308 nm at 20 min, indicating a denser lamellar structure, before steadily increasing to 528 nm at 120 min. This increase in ILS reflects the ongoing dissolution of cementite particles and the subsequent increase in the spacing between the lamellae.

## 4. Discussion

The focus of this research revolves around the effect of chromium (Cr) and molybdenum (Mo) alloying elements on the morphology of cementite lamellae within pearlitic steel. The in-depth exploration of these effects, underpinned by both empirical investigations and numerical simulations, clarifies the pivotal role of these alloying elements in the transformation dynamics of austenite to pearlite and the subsequent ramifications on the microstructural properties. The scatter observed in the data related to the morphology of cementite lamellae can be attributed to several factors associated with the microstructure of pearlitic steels. Some of the key reasons contributing to this scatter are as follows.

The morphology of cementite lamellae is not uniform throughout a given steel sample. It exhibits variations in size, shape, and distribution. It can vary significantly depending on the location within the material. To address this, multiple pictures are typically taken and characterized. This approach helps to ensure that the collected data represent a range of morphological attributes present in different regions of the material.Pearlite formation is highly dependent on grain size, grain shape, and orientation. The size and orientation of pearlite colonies can vary on the grain structure of the steel. In many cases, only colonies perpendicular to the viewing direction are recorded and measured. This selective approach is used to maintain consistency in the data, but it can contribute to scatter, especially when colonies in different grain size areas are not considered.Cementite lamellae within pearlite colonies can have a wide distribution of characteristics. Some lamellae may be round, while others may be elongated. To maintain the representativeness of the data, it is common practice to measure all visible lamellae, regardless of their shape. This inclusivity can introduce scatter into the data sets due to the diversity of lamella shapes.

Therefore, to fully understand the scatter in the data, it is crucial to interpret the graphs in conjunction with the corresponding figures provided in Appendix B. Considering this distribution, let us discuss how additions of Cr and Mo to the reference alloy affect pearlite formation and morphology.

### 4.1. Influence of Alloying Elements on Cementite Morphology

The length of cementite lamellae exhibited significant variations in both experimental and simulated data sets when Cr or Mo were introduced separately to the 1C binary alloy. In general, the addition of Cr resulted in longer and thinner lamellae, whereas the addition of Mo resulted in shorter and thicker lamellae (similar to previous reports [70]), whereas certain conditions for 1.4Mo resulted in lengths comparable to 1C. There was a significant decrease in lamellar thickness upon alloying with Cr (previous publications have also reported similar trends [71]) and slight decrease in lamella thickness with the addition of Mo. It is indicated that the lamellae became more equiaxed, or rounder, with the addition of these alloying elements. However, the effects on circularity were not uniform, as the 1C alloy had greater circularity under certain conditions. The observed trends, like the previous observations [72,73], indicate the strong effect of alloying, with a significant dependence on temperature and time. 

### 4.2. Interplay between Thermodynamics and Kinetics

The addition of Cr and Mo atoms to pearlitic steel (1C) can alter the morphology of cementite by modifying phase transformation dynamics during pearlite formation [12,13]. Both Cr and Mo, known for their tendency to form carbides, have a strong affinity for carbon and frequently form carbide compounds [70,71,73]. The formation of these compounds can cut off carbon atoms, thus reducing their availability for the formation of cementite. This results in finer cementite lamellae, as confirmed by both experimental and simulation data. The results suggest that the addition of Cr or Mo to the base alloy (1C) results in cementite lamellae that are smaller (shorter and thinner), rounder (greater circularity), and more densely packed (less spacing between lamellas). Morphological changes appear more pronounced in the simulation data compared to the experimental data.

### 4.3. Comparative Analysis of Simulated and Experimental Data

The previous sections established the notable effects of Cr and Mo additions on the morphological attributes of cementite lamellae. However, a comparison of the simulation predictions and experimental results further elucidates the complexities involved in accurately predicting these microstructural transformations. 

The relative comparison of the simulation predictions with the experimental observations is calculated for each physical quantity of each alloy to analyze the discrepancy between the simulation predictions and the experimental results. For 1C, it is observed that the simulations, on average, overestimate the length, thickness, and interlamellar spacing compared to the experimental data. On the other hand, the circularity of the cementite lamellae is well predicted by the simulations. For the 1.4Cr and 1.4Mo alloys, similar data are observed across morphological attributes. In particular, the simulations tended to overestimate the length, thickness, and interlamella spacing, while underestimating the circularity. The magnitude of these discrepancies was generally higher for the 1.4Mo alloy. The simulation results indicate more pronounced changes in morphology with changes in temperature and time compared to the experimental results. These differences can be attributed to the controlled and simplified conditions in numerical simulations and to the smaller area that is considered.

Some discrepancies observed between the simulation predictions and the experimental results underscore the inherent complexities and challenges associated with simulating phase transformations in alloy systems. Simulation models, although powerful and insightful, are simplified compared to physical systems. These could be related to the thermodynamic and kinetic parameters, the initial and boundary conditions, or the very formulation of the phase field model itself.

## 5. Conclusions

This comprehensive study, combining both experimental investigations and phase field numerical simulations, illuminates the influential role of chromium (Cr) and molybdenum (Mo) in the phase transformations of hypereutectoid steel. On the basis of the systematic analysis of the obtained data and subsequent discussions, the following conclusions can be drawn.

The introduction of 1.4 wt.% Cr and 1.4 wt.% Mo to the base Fe-1C alloy caused significant alterations in pearlite morphology within the hypereutectoid steel. Both Cr and Mo had a profound effect on the morphology of cementite lamellae, resulting in shorter, thinner, and rounder lamellae, with increased density.Cr, with its inherent ability to stabilize the pearlite formation process, promoted a more homogeneous microstructure. This was empirically evident through the observed decrease in the length of cementite lamellae in the 1.4Cr alloy, further reinforcing the role of Cr as a stabilizing agent.On the contrary, the addition of Mo, while reducing data variance, led to changes in pearlite morphology without necessarily enhancing it. This was established through a relatively non-homogeneous length of lamellae in the 1.4Mo alloy, revealing the influence of Mo on pearlite formation.It is observed that the developed phase field numerical model with identified parameters and given methodology can accurately capture the trends observed experimentally, and can be useful in providing a detailed dynamic understanding of why a certain morphology is observed and evolves.From a processing point of view, it can be concluded that adding 1.4 wt.% chromium to Fe-1C can significantly improve the stability and morphology of pearlite while increasing the length of the pearlite nose in the TTT diagram, and hence can make this alloy suitable for production through a variety of processes, like casting, 3D printing, and forming in wider temperature windows while maintaining a consistent fully pearlitic morphology for desired properties.

The experimental and simulation results reinforce the necessity for a synergistic approach in materials science research, validating and refining computational predictions with experimental findings. 

## Figures and Tables

**Figure 1 materials-17-03538-f001:**
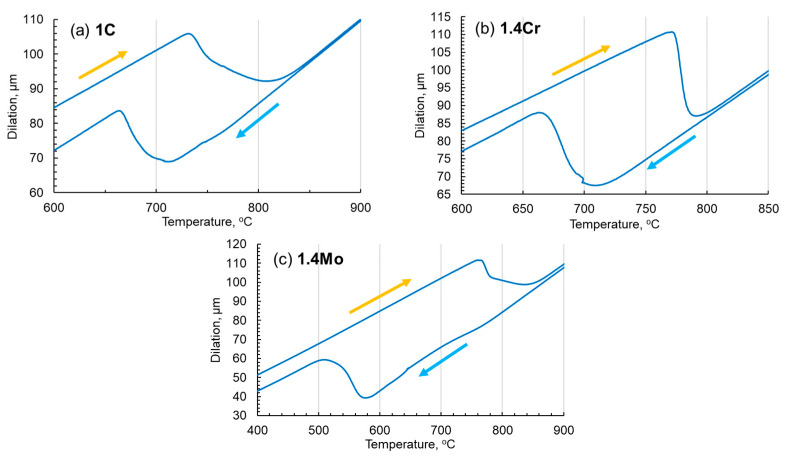
The temperature dilation curves of (**a**) 1C, (**b**) 1.4Cr, and (**c**) 1.4Mo, at heating and cooling rates of 1 °C/s. These graphs show critical temperatures of phase transformations in three alloys during heating and cooling with changing sample elongations. The yellow arrows represent the heating side, whereas the blue arrows represent the cooling side. It is evident from these curves that, with the addition of chromium and molybdenum (phase stability elements), the gap between phase transformation temperatures during heating and cooling increases. Note that the temperature and dilation scales for each alloy are adjusted to clearly represent the transformation windows. The small kink around 700 °C in (**b**) probably appeared due to the slight slippage of the sample between the holding arms due to shrinkage during cooling.

**Figure 2 materials-17-03538-f002:**
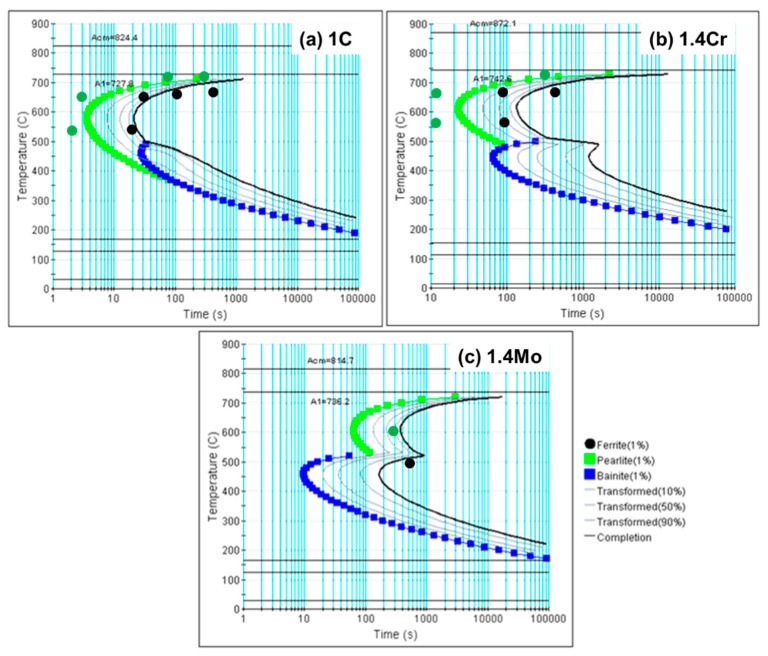
TTT diagrams of (**a**) 1C, (**b**) 1.4Cr, and (**c**) 1.4Mo, calculated using JMatPro v12. The effect of adding chromium and molybdenum to Fe-1C on changing transformation profiles is clearly visible. The round dots represent experimental observations, green dots represent the start of the transformation, and black dots signify the end of the transformation. Please note that the experimental dots are estimated from the time–temperature–tranformation data (without observing the microstructure at exactly these points).

**Figure 3 materials-17-03538-f003:**
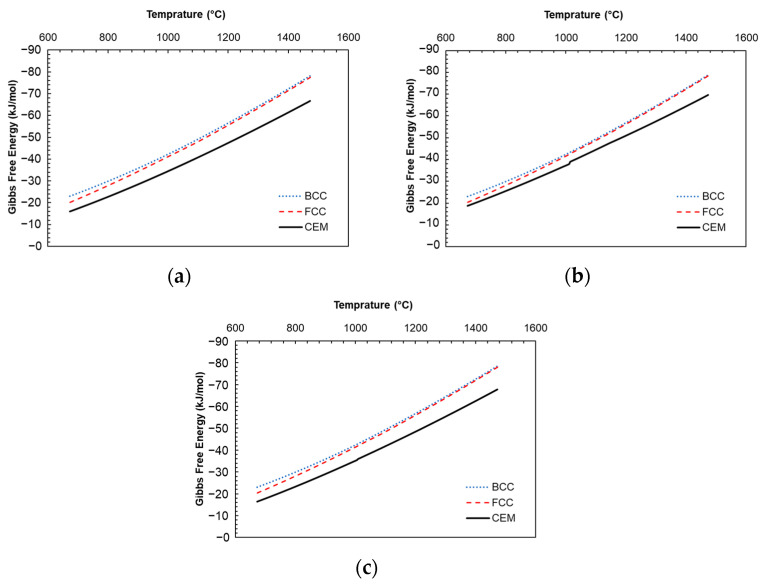
Change in Gibbs free energy based on temperatures in phases for materials (**a**) 1C, (**b**) 1.4Cr, and (**c**) 1.4Mo, where ‘fcc’ is the austenite phase (red dashed line), ‘bcc’ is the ferrite phase (blue dotted line), and ‘cem’ is the cementite phase (black solid line). The spikes in (**b**) are due to the crystal structure change from fcc to bcc and are reflected in the Gibbs free energy of the cementite phase, which becomes more stable below this temperature. The spikes are present but significantly less visible in the other two alloys.

**Figure 4 materials-17-03538-f004:**
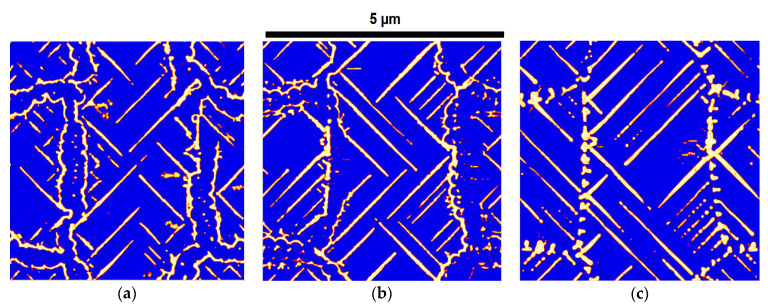
The influence of $M_{\alpha \gamma }$ on the predicted microstructures with three different values: (**a**) 5.0 × $10^{-6}$, (**b**) 5.0 × $10^{-7}$, and (**c**) 5.0 × $10^{-8}$. Here, $M_{\gamma \theta }$ and $M_{\alpha \theta }$ are similar in all three simulations (9.0 × $10^{-7}$ and 1.0 × $10^{-8}$, respectively). The units of the mobility parameters are ${cm}^{4}J^{-1}s^{-1}$. Here the blue color represent ferrite, white represents cementite and red represents the interface of two phases.

**Figure 5 materials-17-03538-f005:**
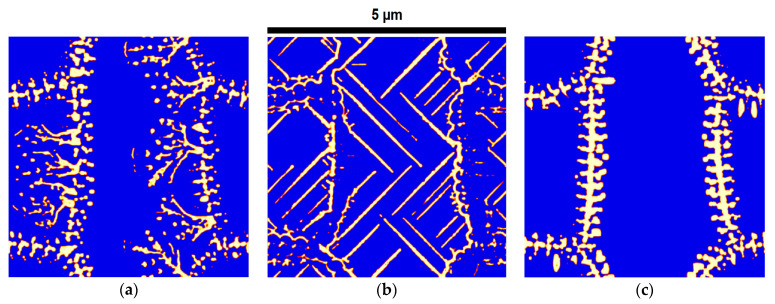
The influence of $M_{\gamma \theta }$ on predicted microstructures under three different values: (**a**) 9.0 × $10^{-6}$, (**b**) 9.0 × $10^{-7}$, and (**c**) 9.0 × $10^{-8}$. Here, $M_{\alpha \gamma }$ and $M_{\alpha \theta }$ are similar in all three simulations (5.0 × $10^{-7}$ and 1.0 × $10^{-8}$, respectively). The units of the mobility parameters are ${cm}^{4}J^{-1}s^{-1}$. Here the blue color represent ferrite, white represents cementite and red represents the interface of two phases.

**Figure 6 materials-17-03538-f006:**
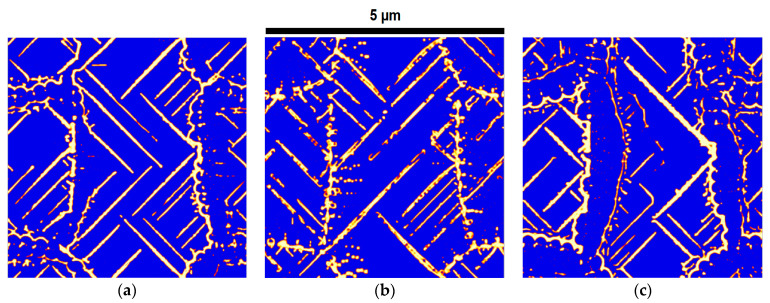
The influence of $M_{\alpha \theta }$ on predicted microstructures under three different values: (**a**) 1.0 × $10^{-7}$, (**b**) 1.0 × $10^{-8}$, and (**c**) 1.0 × $10^{-9}$. Here, $M_{\alpha \gamma }$ and $M_{\gamma \theta }$ are similar in all three simulations (5.0 × $10^{-7}$ and 9.0 × $10^{-7}$, respectively). The units of the mobility parameters are ${cm}^{4}J^{-1}s^{-1}$. Here the blue color represent ferrite, white represents cementite and red represents the interface of two phases.

**Figure 7 materials-17-03538-f007:**
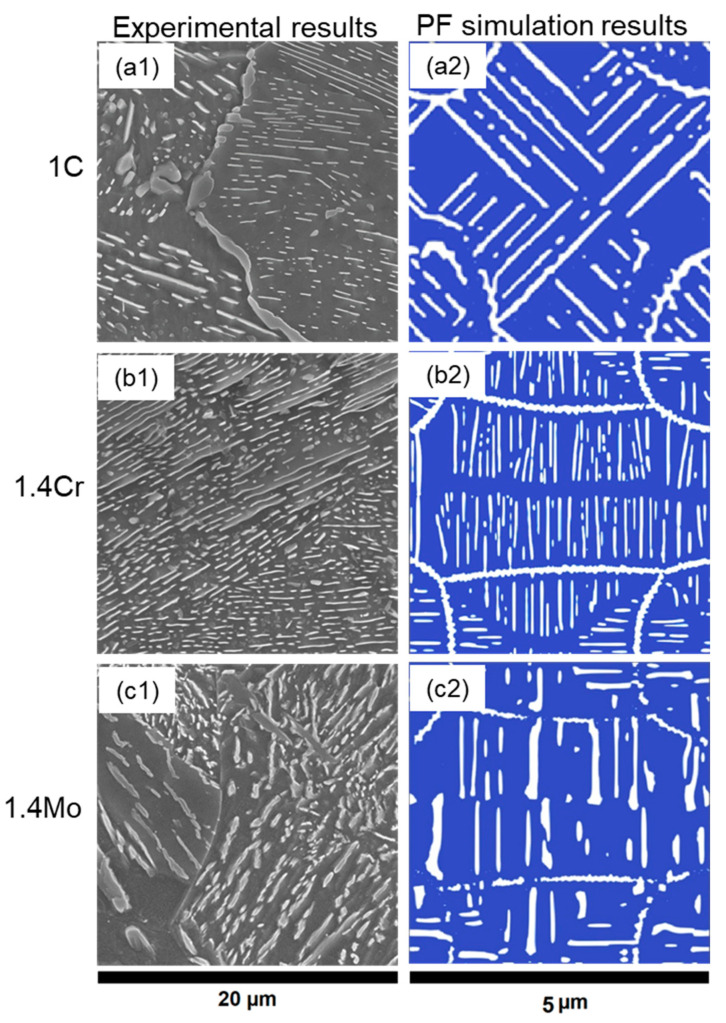
Comparison of experimentally obtained micrographs, where light grey represents cementite lamella and dark grey represents ferrite (**1**) with observed simulation results, where white represents cementite laths (**2**) for validation of the calibrated PFM simulation fitting parameters of (**a1**,**a2**) 1C, (**b1**,**b2**) 1.4Cr, and (**c1**,**c2**) 1.4Mo.

**Figure 8 materials-17-03538-f008:**
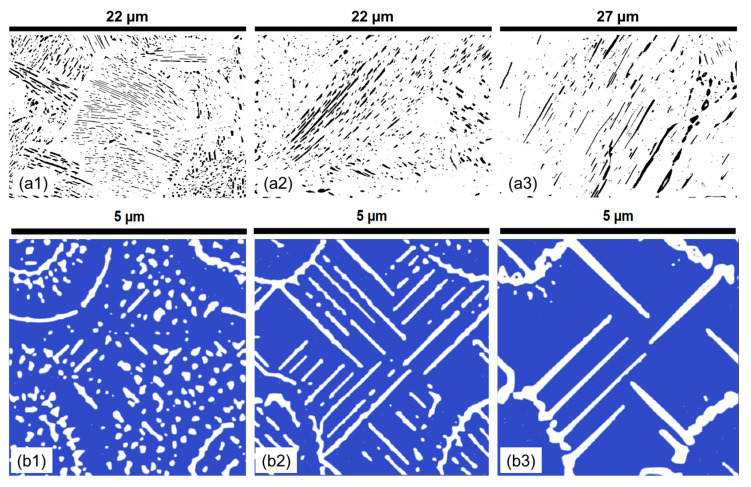
The experimental pearlite morphology obtained after post-processing SEM images compared with numerical predictions in the 1C alloy at varying temperatures for a 30 min holding time. (**a1**) 580 °C; (**a2**) 615 °C; (**a3**) 650 °C; (**b1**) 530 °C; (**b2**) 580 °C; (**b3**) 630 °C. In all experimental subfigures (**a1**–**a3**) black represents the cementite lamella and white represents the ferrite. In all PFM simulations subfigures (**b1**–**b3**) while represents cementite and blue represents ferrite.

**Figure 9 materials-17-03538-f009:**
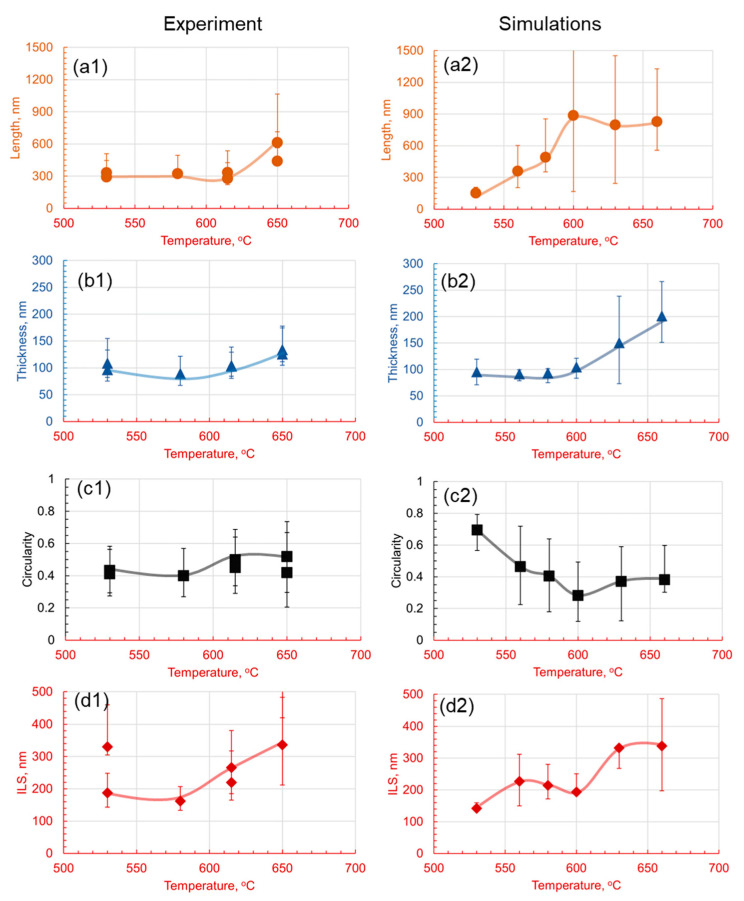
Effect of the holding temperature on the morphology of pearlite at 1C for a 30 min holding time. Subfigures (**1**) are experimental results and subfigures (**2**) are PFM simulations results. The comparison between experimental results and numerical predictions is drawn between the (**a1**,**a2**) length, (**b1**,**b2**) thickness, (**c1**,**c2**) circularity, and (**d1**,**d2**) interlamellar spacing.

**Figure 10 materials-17-03538-f010:**
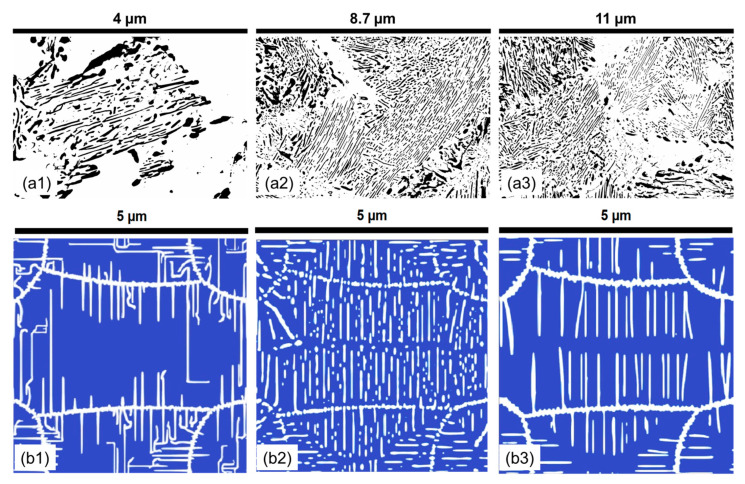
The experimental pearlite morphology obtained after post-processing SEM images compared with numerical predictions in the 1.4Cr alloy at varying temperatures for a 60 min holding time. (**a1**) 560 °C; (**a2**) 610 °C; (**a3**) 650 °C; (**b1**) 560 °C; (**b2**) 610 °C; (**b2**) 690 °C. In all experimental subfigures (**a1**–**a3**) black represents the cementite lamella and white represents the ferrite. In all PFM simulations subfigures (**b1**–**b3**) while represents cementite and blue represents ferrite.

**Figure 11 materials-17-03538-f011:**
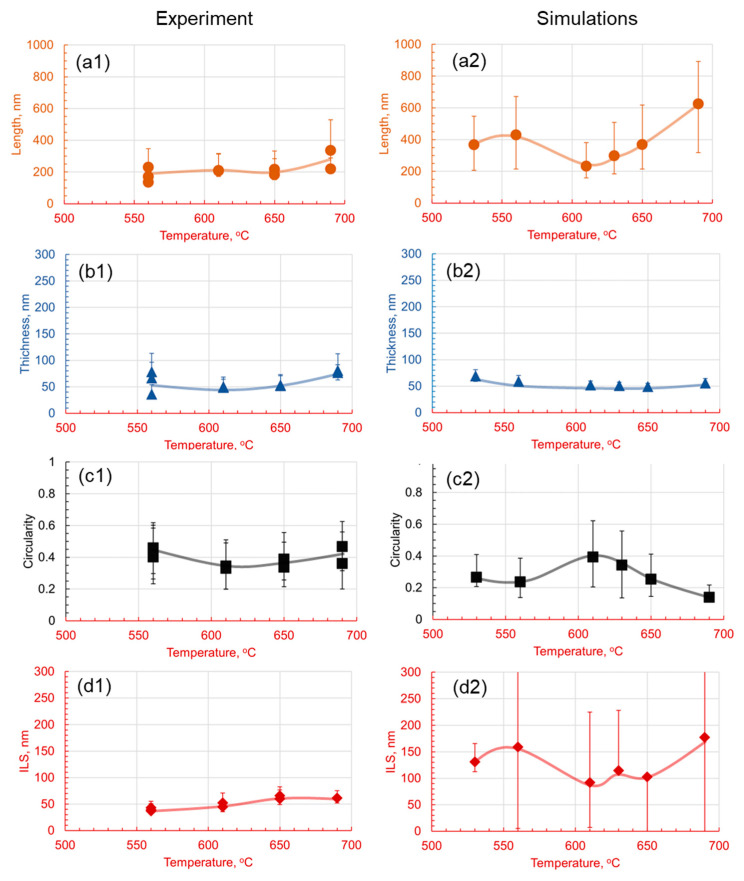
Effect of the holding temperature on the morphology of pearlite in 1.4Cr for a 60 min holding time. Subfigures (**1**) are experimental results and subfigures (**2**) are PFM simulations results. The comparison between experimental results and numerical predictions is drawn between the (**a1**,**a2**) length, (**b1**,**b2**) thickness, (**c1**,**c2**) circularity, and (**d1**,**d2**) interlamellar spacing.

**Figure 12 materials-17-03538-f012:**
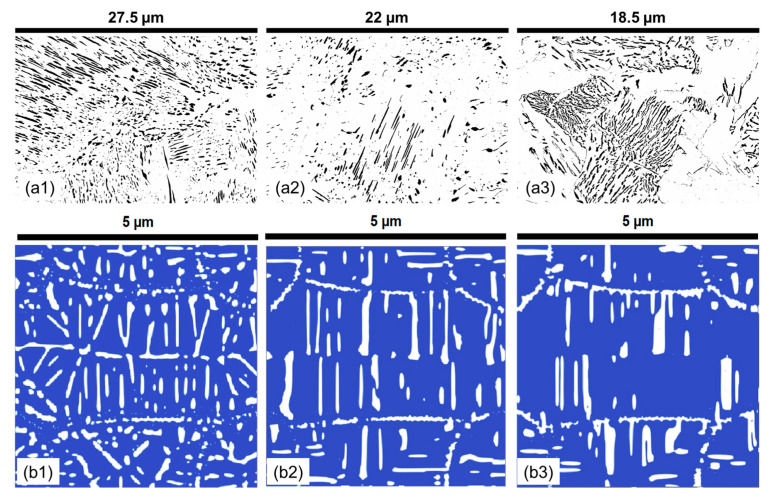
The experimental pearlite morphology obtained after post-processing SEM images compared with numerical predictions in the 1.4Mo alloy at varying temperatures for a holding time of 90 min. (**a1**) 550 °C; (**a2**) 590 °C; (**a3**) 635 °C; (**b1**) 500 °C; (**b2**) 550 °C; (**b3**) 660 °C. In all experimental subfigures (**a1**–**a3**) black represents the cementite lamella and white represents the ferrite. In all PFM simulations subfigures (**b1**–**b3**) while represents cementite and blue represents ferrite.

**Figure 13 materials-17-03538-f013:**
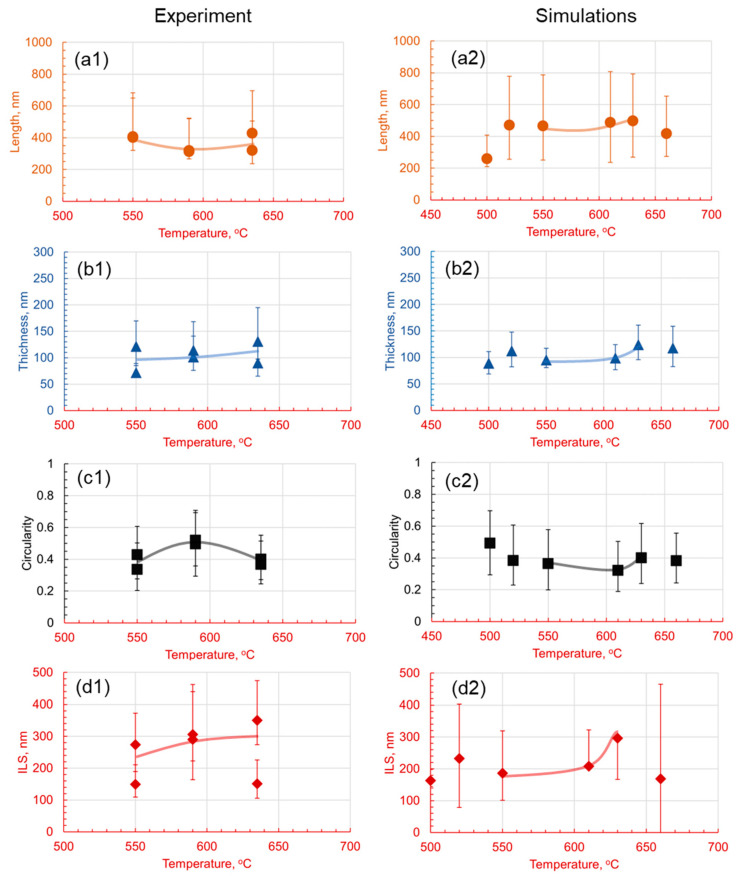
Effect of the holding temperature on pearlite morphology in 1.4Mo for a 90 min holding time. Subfigures (**1**) are experimental results and subfigures (**2**) are PFM simulations results. The comparison between experimental results and numerical predictions is drawn between the (**a1**,**a2**) length, (**b1**,**b2**) thickness, (**c1**,**c2**) circularity, and (**d1**,**d2**) interlamellar spacing.

**Figure 14 materials-17-03538-f014:**
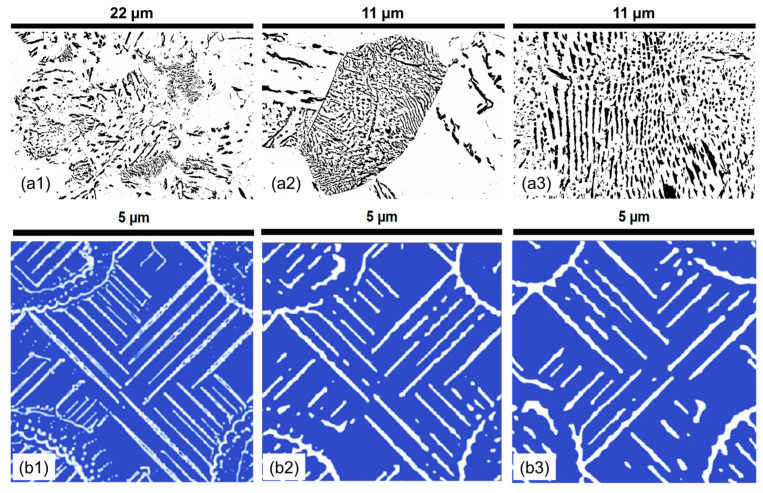
The experimental pearlite morphology obtained after post-processing SEM images compared with numerical predictions in the 1C alloy at varying holding times at a holding temperature of 580 °C. (**a1**) 1 min; (**a2**) 3 min; (**a3**) 5 min; (**b1**) 5 min; (**b2**) 30 min; (**b3**) 60 min. In all experimental subfigures (**a1**–**a3**) black represents the cementite lamella and white represents the ferrite. In all PFM simulations subfigures (**b1**–**b3**) while represents cementite and blue represents ferrite.

**Figure 15 materials-17-03538-f015:**
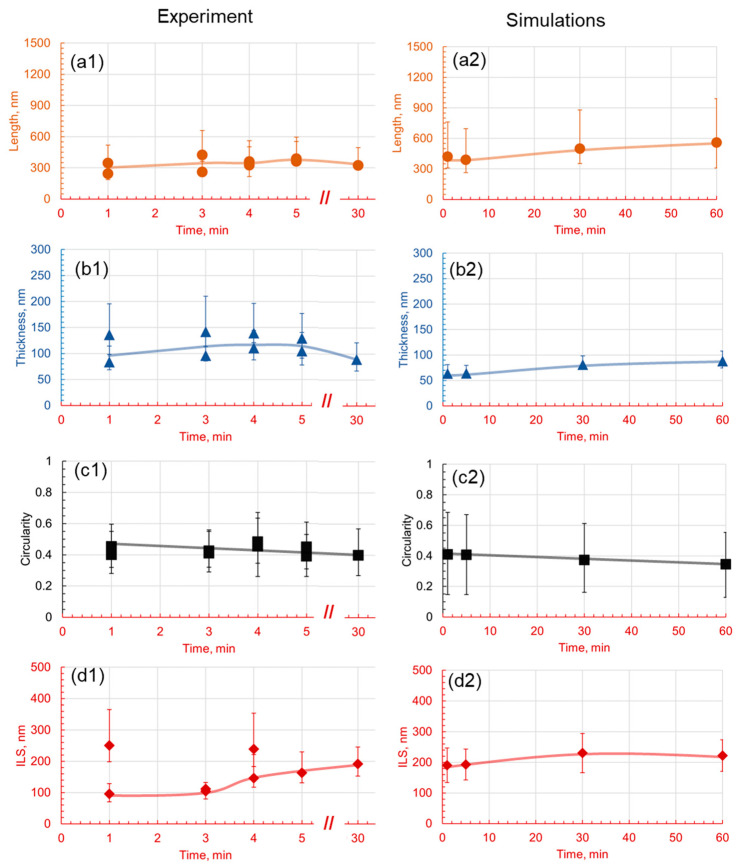
Effect of holding time on pearlite morphology at 1C at a 580 °C holding temperature. Subfigures (**1**) are experimental results and subfigures (**2**) are PFM simulations results. The comparison between experimental results and numerical predictions is drawn between the (**a1**,**a2**) length, (**b1**,**b2**) thickness, (**c1**,**c2**) circularity, and (**d1**,**d2**) interlamellar spacing.

**Figure 16 materials-17-03538-f016:**
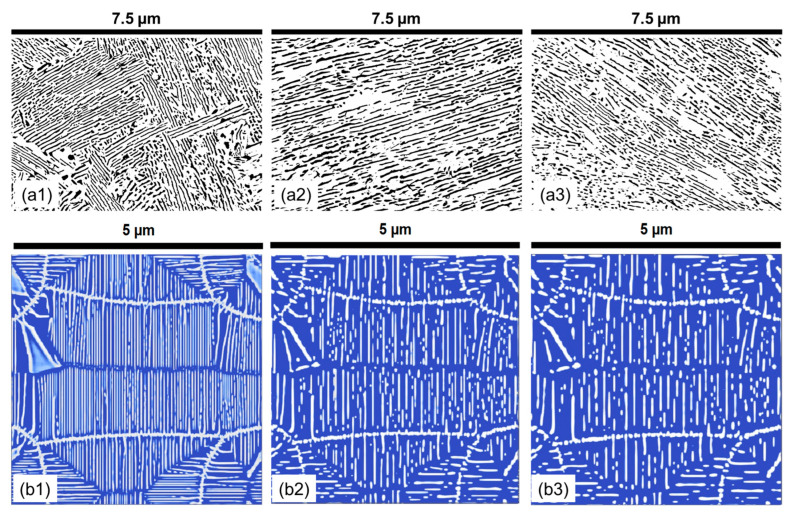
The experimental pearlite morphology obtained after post-processing the SEM images compared with numerical predictions in the 1.4Cr alloy at varying holding times at a holding temperature of 650 °C. (**a1**) 5 min; (**a2**) 10 min; (**a3**) 15 min; (**b1**) 5 min; (**b2**) 30 min; (**b3**) 60 min. In all experimental subfigures (**a1**–**a3**) black represents the cementite lamella and white represents the ferrite. In all PFM simulations subfigures (**b1**–**b3**) while represents cementite and blue represents ferrite.

**Figure 17 materials-17-03538-f017:**
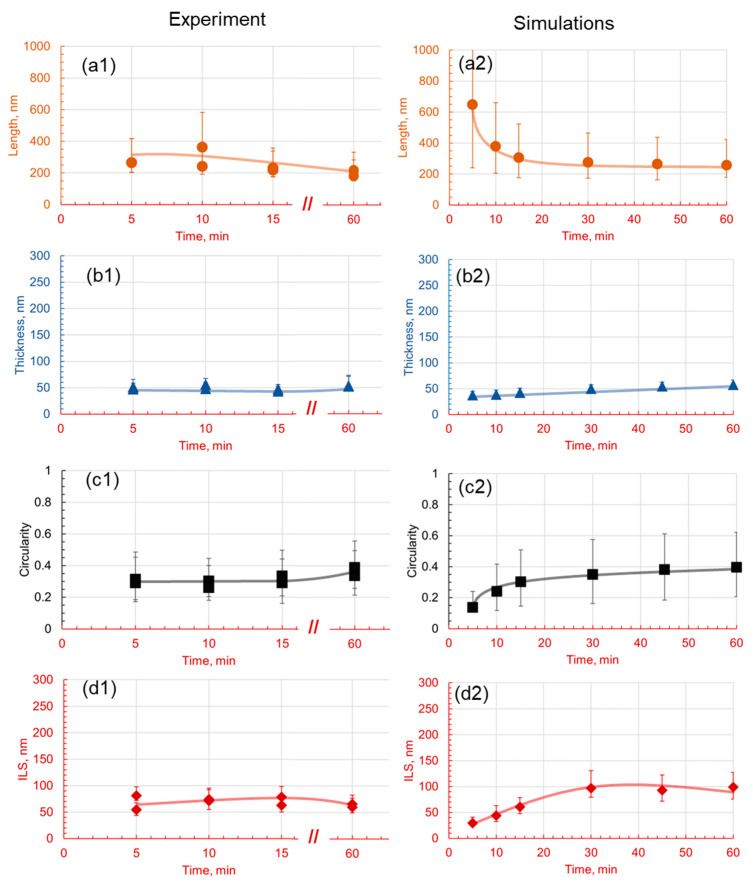
Effect of holding time on pearlite morphology in alloy 1.4Cr at a 650 °C holding temperature. Subfigures (**1**) are experimental results and subfigures (**2**) are PFM simulations results. The comparison between experimental results and numerical predictions is drawn between the (**a1**,**a2**) length, (**b1**,**b2**) thickness, (**c1**,**c2**) circularity, and (**d1**,**d2**) interlamellar spacing.

**Figure 18 materials-17-03538-f018:**
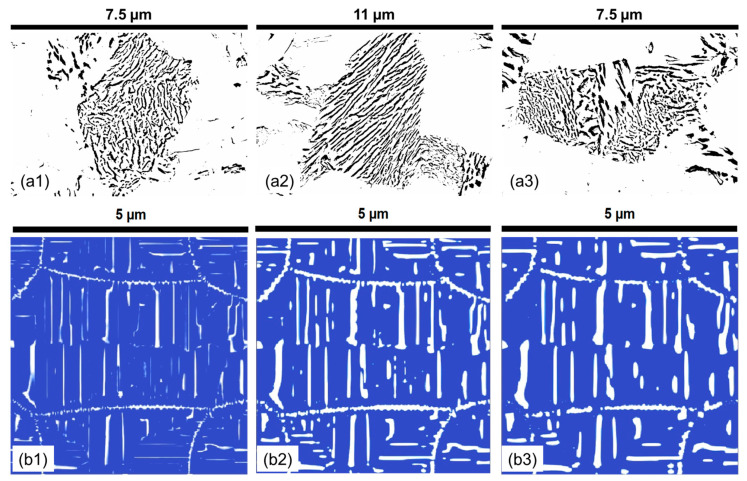
The experimental pearlite morphology obtained after post-processing the SEM images compared with numerical predictions in the 1.4Mo alloy at varying holding times at a holding temperature of 550 °C. (**a1**) 10 min; (**a2**) 20 min; (**a3**) 30 min; (**b1**) 10 min; (**b2**) 30 min; (**b3**) 90 min. In all experimental subfigures (**a1**–**a3**) black represents the cementite lamella and white represents the ferrite. In all PFM simulations subfigures (**b1**–**b3**) while represents cementite and blue represents ferrite.

**Figure 19 materials-17-03538-f019:**
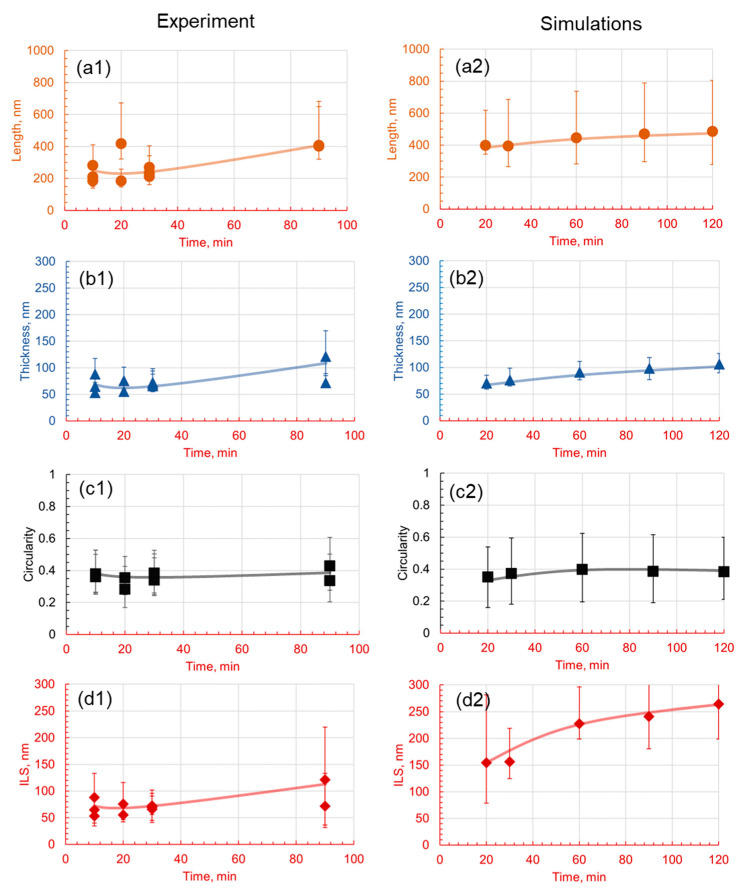
Effect of the holding time on pearlite morphology in 1.4Mo at a holding temperature of 550 °C. Subfigures (**1**) are experimental results and subfigures (**2**) are PFM simulations results. The comparison between experimental results and numerical predictions is drawn between the (**a1**,**a2**) length, (**b1**,**b2**) thickness, (**c1**,**c2**) circularity, and (**d1**,**d2**) interlamellar spacing.

**Table 1 materials-17-03538-t001:** Chemical composition of the casted alloys with appropriate acronyms used throughout the document for reference.

Alloy Name	C	Cr	Mo	Si	Mn	Fe	Acronym
Fe-1C	1.01	0.003	0.005	0.005	0.016	Bal.	1C
Fe-1C-1.4Cr	0.97	1.42	0.005	0.005	0.02	Bal.	1.4Cr
Fe-1C-1.4Mo	0.96	0.01	1.4	0.005	0.02	Bal.	1.4Mo

**Table 2 materials-17-03538-t002:** Identified phase transformation temperatures for 1C, 1.4Cr, and 1.4Mo from the dilatometry test data presented in Figure 1. The beginning and finishing of the respective phase transformation temperatures are represented by subscripts -B and -F, respectively. The values without subscripts are the mean values. All reported temperatures are in °C.

	A_C1-B_	A_C1-F_	A_C1_	A_C3-B_	A_C3-F_	A_CM_
1C	730	738	734	790	831	810.5
1.4Cr	768	776	772	781	801	791
1.4Mo	755	769	762	822	860	841
	A_R1-B_	A_R1-F_	A_R1_	A_R3-B_	A_R3-F_	A_R3_
1C	667	659	663	725	689	707
1.4Cr	675	659	667	733	689	711
1.4Mo	546	492	519	590	563	576.5

**Table 3 materials-17-03538-t003:** The thermodynamic parameters used in this study for the three materials. Here, α is ferrite, γ corresponds to austenite, and θ corresponds to the cementite phase.

Type		Phase	1C	1.4Cr	1.4Mo
Interface	$\mathrm{Interfacial}\;\mathrm{energy},\;J\;{cm}^{-2}$ [59,66,67]	α/α	$7.6\times 10^{-5}$	$7.6\times 10^{-5}$	$7.6\times 10^{-5}$
		α/γ	$7.2\times 10^{-5}$	$7.2\times 10^{-5}$	$7.2\times 10^{-5}$
		α/θ	$7.1\times 10^{-5}$	$7.1\times 10^{-5}$	$7.1\times 10^{-5}$
		γ/γ	$7.6\times 10^{-5}$	$7.6\times 10^{-5}$	$7.6\times 10^{-5}$
		γ/θ	$6.7\times 10^{-5}$	$6.7\times 10^{-5}$	$6.7\times 10^{-5}$
	Mobility, ${cm}^{4}J^{-1}s^{-1}$ [66]	α/α	$3.5\times 10^{-6}$	$3.5\times 10^{-6}$	$3.5\times 10^{-6}$
		α/γ	Variable	Variable	Variable
		α/θ	Variable	Variable	Variable
		γ/γ	$5.0\times 10^{-8}$	$5.0\times 10^{-8}$	$5.0\times 10^{-8}$
		γ/θ	Variable	Variable	Variable
Carbon Diffusion	$D_{0}$$,\;cm^{2}s^{-1}$ [68,69]	γ	$1.5\times 10^{-1}$		
		α	$22.0\times 10^{-1}$		
		θ	$0.18\times 10^{-1}$		
	$Q,\;KJ\;{mol}^{-1}$ [68,69]	α	$1.421\times 10^{2}$		
		γ	$1.225\times 10^{2}$		
		θ	$1.728\times 10^{2}$		

**Table 4 materials-17-03538-t004:** The calibrated prefactor mobility between phases for all three materials. The units are ${cm}^{4}J^{-1}s^{-1}$.

Selected Mobility Parameters	1C	1.4Cr	1.4Mo
$M_{\alpha \gamma }$	$5.0\;\times \;10^{-7}$	$1.0\;\times \;10^{-7}$	$1.0\;\times \;10^{-6}$
$M_{\gamma \theta }$	$9.0\times 10^{-7}$	$1.0\;\times \;10^{-6}$	$5.0\;\times \;10^{-6}$
$M_{\alpha \theta }$	$1.0\;\times \;10^{-8}$	$5.0\;\times \;10^{-9}$	$1.0\;\times \;10^{-8}$

## Data Availability

The SEM images and their corresponding distribution plots are included in the Appendix A and Appendix B; further inquiries can be directed to the corresponding authors. The scrips and more data supporting the conclusions of this article will be available from the authors upon request.

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
