# Peer review of "Analyzing the Effects of Cr and Mo on the Pearlite Formation in Hypereutectoid Steel Using Experiments and Phase Field Numerical Simulations"

_materials, 2024, doi:10.3390/ma17143538_

Round 1

Reviewer 1 Report

Comments and Suggestions for Authors

The current paper reports on the ‘Analyzing the effect of Cr and Mo on the pearlite formation in hypereutectoid steel based on experiments and phase field numerical simulations’. The paper is well structured, well organised and the presented claims are supported by experimental proofs. The only criticism is that, the introduction section needs to enhance in terms of critical analysis of the available information in this field and highlight the novelty of the present work. Based on my assessment, I suggest minor revision of this paper.

The detail comments are as follows:

1.      The abstract must be re-written in a concise form. Instead of experimental details, it should contain the main essence of the present work including major findings, if possible in terms of numerical data.

2.      Line 52-74: Can be avoided and not relevant. These types of experimental technique are widely used and there is no need to repeat it here.

3.      The introduction section may be better presented in terms of critical analysis of the current findings in this filed. It current form it looks the introduction of a book chapter, not a journal paper.

4.      Again, the last paragraph of the introduction section can be omitted, as it is a journal paper not a book chapter.

5.      Please highlight the research gap/novelty of the present work.

6.      In current form, the introduction section is too descriptive without any critical analysis of previous in this field. Thus, the whole introduction section needs to be revised.

7.      Fig. 7 and so on: The scale-bar is missing in the micrographs.

8.      Fig. 8a1,a2,a3: Are those optical image or electron microscopy image? Should be indicated clearly.

9.      Line 371-375: How the thickness was measured? From the SEM micrographs? Should be mentioned clearly.

Comments on the Quality of English Language

Minor editing of English language required

Author Response

Response to Reviewer 1 comments

General Comment: The current paper reports on ‘Analyzing the effect of Cr and Mo on pearlite formation in hypereutectoid steel based on experiments and phase field numerical simulations’. The paper is well structured and well organized, and the presented claims are supported by experimental proofs. The only criticism is that the introduction section needs to enhance in terms of critical analysis of the available information in this field and highlight the novelty of the present work. On the basis of my assessment, I suggest a minor revision of this paper.

General Response: The authors thank the reviewers for taking the time to review the article and providing valuable feedback, which helped us improve clarity and outlook. All reviewer comments have been appropriately addressed in the main body of the article, and the modified text is highlighted in yellow for easier comparison. The article was read thoroughly to check for spelling or punctuation mistakes and to improve the language professionally. The introduction of the article has been improved by removing paragraph 2 and 3 that were more redundant, by adding more recent work, related to the field of work and by highlighting the novelty of the current work more appropriately.

Point-by-point response to each reviewer's comment with reason and subsequent modification in the article body is presented below.

Comment 1: The abstract must be rewritten in a concise form. Instead of experimental details, it should contain the main essence of the present work, including major findings, if possible in terms of numerical data.

Response 1: The abstract has not been revised thoroughly, it now reads as:

‘In this study, we quantitatively investigate the impact of 1.4 wt. % chromium and 1.4 wt. % molybdenum additions on pearlitic microstructure characteristics in 1 wt. % carbon steels. The study was carried out using a combination of experimental methods and phase field simulations. We utilized MatCalc and JMatPro to predict transformation behaviors and electron microscopy for microstructural examination, focusing on pearlite morphology under varying thermal conditions. Phase-field simulations were carried out using MICRESS software and, informed by thermodynamic data from MatCalc and literature, were conducted to replicate pearlite formation, in good agreement with experimental observations. In this work we introduced a semi-automatic reliable microstructural analysis method, quantifying features like lamella dimensions and spacing through image processing by Fiji ImageJ. The introduction of Cr resulted in longer, thinner, and more homogeneously distributed cementite lamellae, while Mo led to shorter, thicker lamellae. Phase-field simulations accurately predicted these trends and showed that alloying with Cr or Mo increases the density and circularity of the lamellae. Our results demonstrate that Cr stabilizes pearlite formation, promoting a uniform microstructure, whereas Mo affects the morphology without enhancing homogeneity. The phase field model, validated by experimental data, provides insights into the morphological changes induced by these alloying elements, supporting the optimization of steel processing conditions.’

Comment 2: Lines 52-74: Can be avoided and not relevant. These types of experimental technique are widely used and there is no need to repeat it here.

Response 2: These two paragraphs have been removed.

Comment 3: The introduction section may be better presented in terms of critical analysis of the current findings in this field. It current form looks the introduction of a book chapter, not a journal paper.

Response 3: We agree, the idea was to provide as much details as possible for even new readers to catch up with the work, but now we have removed all the redundant details and have only kept the necessary information in the introduction.

Comment 4: Again, the last paragraph of the introduction section can be omitted, as it is a journal paper, not a book chapter.

Response 4: The last paragraph has now been removed.

Comment 5: Highlight the research gap/novelty of the present work.

Response 5: The gap has now been revised to write more clearly; It is present on Page 3, lines 120-124 and read as:

‘Existing models require labor intensive experimental procedures for calibration, limiting their efficiency and commercial applicability [56, 57]. Furthermore, the use of software tools such as MatCalc to derive thermodynamic parameters from the relevant literature, which is an important step in these simulations, has yet to be properly used. ‘

The novelty and contributions of the current work are written in the last paragraph of the introduction section, which reads:

“To address this, the study sets three primary objectives. First, the objective is to develop a numerical simulation method capable of dynamically predicting the evolution of pearlite morphology and the resultant mechanical behavior in carbon steels. This involves an understanding of how alloying elements such as Cr and Mo influence the pearlitic transformation during the cooling process. Second, the study will use experimental measurements to obtain critical material data and calibrate interface fitting parameters, thereby enhancing the reliability of the simulation model. Third, develop and use a robust microstructural feature characterization technique to quantitatively analyze and compare pearlite morphology across different alloys and heat treatment procedures from experimental results and numerical simulation predictions. This three-pronged approach aims to establish a more efficient methodology to predict and optimizing heat treatment processes in various types of steel. ‘

Comment 6: In current form, the introduction section is too descriptive without any critical analysis of previous work in this field. Thus, the whole Introduction section needs to be revised.

Response 6: Yes, the suggestion is well received, and the introduction section is now thoroughly revised.

Comment 7: Fig. 7 and so on: The scale bar is missing in the micrographs.

Response 7: The scale bars for all figures are provided above them. The simulation results have singular scale, and therefore only one bar is shown for all three figures to avoid repetition.

Comment 8: Fig. 8a1,a2,a3: Are these optical image or electron microscopy images? The reason should be clearly indicated.

Response 8: All these are SEM images; it is now clearly indicated in the figure captions where applicable.

Comment 9: Lines 371-375: How was the thickness measured? From the SEM micrographs? It should be mentioned clearly.

Response 9: The method of lamella morphology measurement is extensively explained in Appendix A. It has been adequately cited at respective positions in the main article body. It was not discussed in the main article body to avoid detouring from the main story and keep the main articles within reasonable length.

Comment 10: Minor editing of English language required.

Response 10: The article has now been thoroughly reviewed to make any spelling or punctuation errors.

Reviewer 2 Report

Comments and Suggestions for Authors

The authors in their work titled "Analyzing the effect of Cr and Mo on the pearlite formation in hypereutectoid steel based on experiments and phase field numerical simulations" presented a very interesting combination of experimental data as well as computer simulations. The entire article is presented in a very good way and will certainly be of interest to the readers of your publishing house. Despite the lack of objections to the substantive part of the article, I have some reservations regarding the presentation of certain results and a few purely editorial comments:

1. The units of the variables used in the equations are not presented throughout the article.

2. Incorrect table numbering was used - the numbering should be presented using Arabic numerals, not Roman ones. Additionally, the table number should be followed by a dot, not a colon (please do the same for drawings).

3. Please standardize the temperature units - the authors alternately use degrees Celsius and degrees Kelvin. Please select one temperature unit.

4. In lines 17 to 172, the authors described the parameters of rolling the samples, but did not provide the manufacturer or parameters of the devices used in this process. Please fill in this gap.

5. Please explain the kink in the lower curve (temperature 700 degrees Celsius) shown in Figure 1(b).

6. Please align all equations to the text of the article (on the left).

7. All units presented in charts or tables should be preceded by a comma and not written in brackets.

8. Why does the Gibbs free enthalpy curve in Figure 3(b) show a sudden spike in values ​​at just above 1000 degrees Celsius? Please explain this phenomenon.

9. Table 3 has been divided into two parts - please correct it

Author Response

Response to Reviewer 2 comments

General comment: The authors in their work titled "Analyzing the effect of Cr and Mo on the pearlite formation in hypereutectoid steel based on experiments and phase field numerical simulations" presented a very interesting combination of experimental data as well as computer simulations. The entire article is presented in a very good way and will certainly be of interest to the readers of your publishing house. Despite the lack of objections to the substantive part of the article, I have some reservations regarding the presentation of certain results and a few purely editorial comments:

General Response: The authors thank the reviewers for taking the time to review the article and providing valuable feedback, which helped us improve clarity and outlook. All reviewer comments have been appropriately addressed in the main body of the article, and the modified text is highlighted in yellow for easier comparison. The article was read thoroughly to check for spelling or punctuation mistakes and to improve the language professionally. The introduction of the article has been improved by removing paragraph 2 and 3 that were more redundant, by adding more recent work, related to the field of work and by highlighting the novelty of the current work more appropriately.

Point-by-point response to each reviewer's comment with reason and subsequent modification in the article body is presented below.

Comment 1: The units of the variables used in the equations are not presented throughout the article.

Response 1: The units have been checked thoroughly and now are provided for all the values that are reported.

Comment 2: Incorrect table numbering was used - the numbering should be presented using Arabic numerals, not Roman ones. Additionally, the table number should be followed by a dot, not a colon (please do the same for drawings).

Response 2: Arabic numerals are now used to number tables throughout the document. Also, the table and figure numbers are now followed by a dot instead of a colon.

Comment 3: Please standardize the temperature units - the authors alternately use degrees Celsius and degrees Kelvin. Please select one temperature unit.

Response 3: Thank you for pointing this issue out, the temperature units have now been revised to degrees Celsius throughout the document.

Comment 4: In lines 17 to 172, the authors described the parameters of rolling the samples, but did not provide the manufacturer or parameters of the devices used in this process. Please fill in this gap.

Response 4: The details about the rolling setup are now provided in section 2.1 and read as:

“The cast billets rolled into wide by using multipass rolling starting at 1250 °C and ending at ~850 °C. The rolling experiments were conducted using a Trio-Walzgerüst, a versatile rolling mill designed to accommodate various groove sequences and commonly used for fundamental studies on material flow during caliber rolling. The device has roll diameters ranging from 280 to 320 mm, has maximum rolling steep of 8 m/sec with maximum initial block dimensions of 48 × 48 × 2500 mm, and can produce wires of Φ 6 to 16 mm. The Trio was used to roll 12 mm diameter wire that was then cut into manageable lengths and homogenized at 1250 ° C for 8 hours. Solid and hollow cylindrical samples with 10 mm length and 5 mm outer diameter were manufactured from all reference materials for the dilatometer tests.”

Comment 5: Please explain the kink in the lower curve (temperature 700 degrees Celsius) shown in Figure 1(b).

Response 5: I looked back at the raw data to find a reason for this kink that is visible during cooling. The small kink around 700 ° C in b) probably appeared due to a slight slippage of the sample between the holding arms due to shrinkage during cooling. I have added this information to the figure description to avoid any confusion.

Comment 6: Please align all equations to the text of the article (on the left).

Response 6: All the equations are now aligned with text.

Comment 7: All units presented in charts or tables should be preceded by a comma and not written in brackets.

Response 7: All the units are now preceded by a coma and the brackets have been removed.

Comment 8: Why does the Gibbs free enthalpy curve in Figure 3(b) show a sudden spike in values at just above 1000 degrees Celsius? Please explain this phenomenon.

Response 8: Interestingly it shows a step in all cases. In 1.4Cr case it is most visible. The steps in Gibbs free energy just above 1000°C in all alloys are due to a phase transformation and is highly influenced by the presence of chromium. This transformation results in an abrupt change in the thermodynamic stability of the cementite phase and hence the observed step in the Gibbs free energy curve.

The step is highest in 1.4Cr case because Cr is a stabilizing agent and carbon diffusion difference in fcc and bcc phases of this alloy is highest.

Comment 9: Table 3 has been divided into two parts - please correct it

Response 9: It has now been corrected.

Reviewer 3 Report

Comments and Suggestions for Authors

General review:

Overall, the authors tried to introduce the impact of chromium and molybdenum additions on pearlitic microstructure characteristics in 1 % carbon steels. Employing a combination of experimental and computational approaches, we analyzed the microstructural evolution in benchmark and alloyed steels (Fe-1C, Fe-1C-1.4Cr, Fe-1C-1.4Mo). The experimental phase involved producing 12 mm wires, followed by homogenization and machining to obtain samples for transformation analysis. We utilized MatCalc and JMatPro for predicting transformation behaviors, and electron microscopy for microstructural examination, focusing on pearlite morphology under varying thermal conditions. Phase field simulations, leveraging MICRESS software and informed by thermodynamic data from MatCalc and literature, were conducted to replicate pearlite formation, demonstrating good agreement with experimental observations. We introduced a microstructural analysis method, quantifying features like lamella dimensions and spacing through image processing by Fiji ImageJ. This facilitated a comprehensive database, supporting statistical analyses to uncover the effect of holding temperature and time in reference alloy and how addition of Cr and Mo effect pearlite morphology evolution. This work bridges experimental insights and computational predictions, offering a robust framework for optimizing steel processing by using phase field simulations for choosing the right alloying element with appropriate holding temperature and time to obtain desired microstructure. However, before it can be published, I have some questions about this article and some suggestions:

Minor review:

1. I understood where you want to apply this material introduced in this Introduction section. Then, if you combine the first, second, and third paragraphs in the Introduction Section, it works well, I think.

2. In the Experimental Section, please explain how you can make the predicted microstructures?

3. Overall, the explanation between the predicted microstructures and the real microstructures seemed not to match well each other . Explain about it.

4. Some of the references are too old. Change more recent one.

Author Response

Response to Reviewer 3 comments

General comment: Overall, the authors tried to introduce the impact of chromium and molybdenum additions on pearlitic microstructure characteristics in 1 % carbon steels. Employing a combination of experimental and computational approaches, we analyzed the microstructural evolution in benchmark and alloyed steels (Fe-1C, Fe-1C-1.4Cr, Fe-1C-1.4Mo). The experimental phase involved producing 12 mm wires, followed by homogenization and machining to obtain samples for transformation analysis. We utilized MatCalc and JMatPro for predicting transformation behaviors, and electron microscopy for microstructural examination, focusing on pearlite morphology under varying thermal conditions. Phase field simulations, leveraging MICRESS software and informed by thermodynamic data from MatCalc and literature, were conducted to replicate pearlite formation, demonstrating good agreement with experimental observations. We introduced a microstructural analysis method, quantifying features like lamella dimensions and spacing through image processing by Fiji ImageJ. This facilitated a comprehensive database, supporting statistical analyses to uncover the effect of holding temperature and time in reference alloy and how addition of Cr and Mo effect pearlite morphology evolution. This work bridges experimental insights and computational predictions, offering a robust framework for optimizing steel processing by using phase field simulations for choosing the right alloying element with appropriate holding temperature and time to obtain desired microstructure.

General response: The authors thank the reviewers for taking the time to review the article and providing valuable feedback, which helped us improve clarity and outlook. All reviewer comments have been appropriately addressed in the main body of the article, and the modified text is highlighted in yellow for easier comparison. The article was read thoroughly to check for spelling or punctuation mistakes and to improve the language professionally. The introduction of the article has been improved by removing paragraph 2 and 3 that were more redundant, by adding more recent work, related to the field of work and by highlighting the novelty of the current work more appropriately.

Point-by-point response to each reviewer's comment with reason and subsequent modification in the article body is presented below.

Comment 1: I understood where you want to apply this material introduced in this Introduction section. Then, if you combine the first, second, and third paragraphs in the Introduction Section, it works well, I think.

Response 1: As suggested, the paragraphs 2 and 3 have been mostly removed and paragraph 1 has been reinforced with more recent work to provide information on state of the art.

Comment 2: In the Experimental Section, please explain how you can make the predicted microstructures?

Response 2: I could not understand the comment. I am responding based on the best of my understanding. The experimental methodology has been thoroughly explained in sections 2.1 and 2.2. The methodology of the lamella morphology measurement is explained in detail in Appendix A. I hope this answers the comment.

Comment 3: Overall, the explanation between the predicted microstructures and the real microstructures seemed not to match well each other . Explain about it.

Response 3: Yes, there are some discrepancies between experimentally observed trends and numerical predictions. The similarities and differences have been discussed in detail in Section 4.

Comment 4: Some of the references are too old. Change more recent one.

Response 4: We have now tried to remove some of the redundant old references and have added more recent ones.
